# Enhancing Precision with an Ensemble Generative Adversarial Network for Steel Surface Defect Detectors (EnsGAN-SDD)

**DOI:** 10.3390/s22114257

**Published:** 2022-06-02

**Authors:** Fityanul Akhyar, Elvin Nur Furqon, Chih-Yang Lin

**Affiliations:** 1Department of Electrical Engineering, Yuan Ze University, Taoyuan 320, Taiwan; fityanul@telkomuniversity.ac.id; 2School of Electrical Engineering, Telkom University, Bandung 40257, Indonesia; elvin@novaglobal.com.sg; 3Artificial Intelligence—High Performing Computer Division, Nova Global Pte Ltd., Singapore 609966, Singapore

**Keywords:** defect detection, generative adversarial network (GAN), recursive FPN, boundary localization

## Abstract

Defects are the primary problem affecting steel product quality in the steel industry. The specific challenges in developing detect defectors involve the vagueness and tiny size of defects. To solve these problems, we propose incorporating super-resolution technique, sequential feature pyramid network, and boundary localization. Initially, the ensemble of enhanced super-resolution generative adversarial networks (ESRGAN) was proposed for the preprocessing stage to generate a more detailed contour of the original steel image. Next, in the detector section, the latest state-of-the-art feature pyramid network, known as De-tectoRS, utilized the recursive feature pyramid network technique to extract deeper multi-scale steel features by learning the feedback from the sequential feature pyramid network. Finally, Side-Aware Boundary Localization was used to precisely generate the output prediction of the defect detectors. We named our approach EnsGAN-SDD. Extensive experimental studies showed that the proposed methods improved the defect detector’s performance, which also surpassed the accuracy of state-of-the-art methods. Moreover, the proposed EnsGAN achieved better performance and effectiveness in processing time compared with the original ESRGAN. We believe our innovation could significantly contribute to improved production quality in the steel industry.

## 1. Introduction

Vision-based steel surface inspection system (detectors) are the key to maintaining the production quality of steel products. Even when a high-resolution camera is utilized to acquire images of the steel surface, some difficult features of defects affect the performance of detectors. More specifically, defects that are vague in appearance and tiny may be challenging to detect since they are shown against a wide background area [1]. Therefore, enhancing the quality of input images plays an important role in improving the localization performance of defect detectors. These two problems mean a preprocessing stage, such as recalibrating the pixel quality, is required before the steel image enters the detection stage. The output of this process might increase the difference between the defect and the background area.

### 1.1. The Benefecial of Super Resolution Generative Adversarial Networks

Currently, there are two main options to improve the sharpness of the image via pixel manipulation. Data augmentation is the first most practical way; for example, gamma correction and contrast-limited adaptive histogram equalization (CLAHE) [2]. Although easy to implement, this process increases the noise in the image and reduces the difference between the region of interest and the background. On the other hand, super-resolution is becoming another popular pixel restoration method. In essence, image super-resolution is the process of increasing the resolution of an image from low resolution to high resolution. This method is quite useful in many applications, i.e., surveillance systems [3], medical imaging [4], and industry [5]. Performing facial recognition in security systems requires high-resolution images [6]. Moreover, generating MRI images requires a high resolution to ensure the accuracy of disease diagnosis [7]. The same goes for inspection systems in the industry, where a high resolution can assist with the precision of defect localization in the products [8]. To the best of our knowledge, enhanced super resolution generative adversarial networks (ESRGAN) is a state-of-the-art model for super resolution [9], which is a recent innovation in machine learning. GAN basically employs a deep network in conjunction with an adversarial network to produce higher resolution images. The generated realistic image produced by GAN is very pleasant to human eyes since it adopts the similarity index to measure the distance from the original image. GANs are made up of two networks: the generator and the discriminator. The generator component of a GAN learns to generate synthesis data by incorporating feedback from the discriminator. It learns to make the discriminator classify its output as real. GAN is more applicable to this case since it applies to unseen images in the latest models [10]. Furthermore, multi-generator GAN has significantly improved upon single-generator GAN [11,12]. The basic idea in the multi-generator is to share the network parameters and find the best generator through a probabilistic mechanism. This technique is also applied to real-world applications such as power systems [13], medication systems [14], and transportation systems [11].

### 1.2. The Benefecial of Feature Pyramid Networks 

Feature pyramid networks (FPN) [15] have had greater success in object detection. The FPN structures allow the features to be extracted in different scales, which is the key to solving the challenges for small objects. Furthermore, several works have paid more attention to improving FPN performance. For instance, the proposed work, NAS-FPN [16], used neural architecture search to develop a new feature pyramid architecture in a novel scalable search space that covered all cross-scale connections. The newly-invented architecture uses a combination of top-down and bottom-up connections to connect features at different scales. Moreover, the work in ref. [17] proposed a plugin for feature upsampling in the FPN structure called content-aware reassembly of features (CARAFE) to enlarge the receptive field. Instead of using interpolation and deconvolution, CARAFE employs an adaptive and optimized reassembly kernel in various locations to achieve better performance. Currently, recursive feature pyramid (RFP) [18] extends the functionality of FPN by incorporating extra feedback connections from the FPN layers into the bottom-up backbone layers. The RFP can serve as a backbone for an object detector that looks at the images twice or more by unrolling the recursive structure to a sequential implementation to enhance the generation of a meaningful representation of samples. In line with the successful works on FPN, anchoring methods also play an essential role in object detection. Region proposal network (RPN) [19] is a state-of-the-art anchoring method developed to predict the region proposal with a broader scale and aspect ratio. The anchor boxes proposed are served at multiple scales and aspect ratios in this scheme. The RPN predicts whether an anchor will be in the background or foreground, and refines the anchor accordingly. After discerning the exact box (location) of the foreground or object of interest, the current model [20] proposes the precise localization of the object. This methodology, named Side-Aware Boundary Localization (SABL), estimates an object’s boundary with the bucketing scheme and fine regression. In addition to applying super-resolution GAN, most real-world applications, including manufacturing, benefit from FPN [21,22]. Moreover, the combination of FPN and RPN shows better performance on production in various industries [23,24,25,26].

### 1.3. The Major Contributions and Novelty of the Proposed EnsGAN-SDD 

Motivated by the success of enhanced super-resolution generative adversarial networks (ESRGAN) and feature pyramid networks (FPN) in many real-world applications, this work proposed GAN for the preprocessing stage to improve the quality of steel images before inspection by defect detectors. Moreover, this work aimed to integrate the state-of-the-art ESRGAN in the preprocessing stage and the recursive feature pyramid networks in the detectors stage to develop a new steel surface inspection system. Although GAN is commonly used in industrial applications [27,28,29], to the best of our knowledge, the concept of multi-generator GAN has never previously been applied. Furthermore, in this work, we applied the concept of multi-generator GAN to steel manufacturing by proposing a novel EnsGAN to enhance the output quality and processing speed of conventional ESRGAN. The current multi-generator GAN [12,30,31,32] uses the whole image as the input network. In contrast, the proposed multi-generator in EnsGAN improves upon previous works by splitting operations on the input image, which significantly enhances the processing speed. Meanwhile, we improved defect localization precision by adding the Side-Aware Boundary Localization into the detector’s architecture. Our proposed work, named EnsGAN-SDD, significantly improves the accuracy of the steel surface inspection system compared with state-of-the-art methods like the R-CNN Family [19,33], YOLO variants [34], and transformer networks [35].

## 2. Related Works

Super-resolution generative adversarial network (SRGAN) [36] uses a deep neural network combined with an adversary network to generate higher resolution images. It employs a perceptual loss function that includes an adversarial loss and a content loss. Specifically, a high-resolution image (HR) is downsampled to a low-resolution image (LR) during the training. Then, using a GAN generator, LR images are upsampled to super-resolution images (SR). Finally, it uses a discriminator to distinguish the HR images. Moreover, the backpropagation uses the GAN loss to train the discriminator and generator by employing a perceptual loss function comprising an adversarial loss and a content loss. The adversarial loss directs the solution to the natural image manifold by employing a trained discriminator network to distinguish between super-resolved images and original photo-realistic images. Rather than relying on similarity in pixel space, the proposed work employed a content loss motivated by perceptual similarity. To improve visual quality, this study [9] enhanced three critical components of SRGAN architecture, adversarial loss, and perceptual loss by proposing Enhanced SRGAN (ESRGAN). Specifically, Residual-in-Residual Dense Block (RRDB) without batch normalization was introduced as the basic network building unit. Furthermore, relativistic GAN [37] enables the discriminator to predict relative reality rather than absolute value. Finally, using the features prior to activation, ESRGAN improves perceptual loss and provides stronger supervision for brightness consistency and texture recovery. Furthermore, Real-ESRGAN [10] applies the powerful ESRGAN to a practical restoration application trained on purely synthetic data. Specifically, a high-order degradation modeling process was introduced to simulate complex real-world degradations better. Ringing and overshoot artifacts were also taken into account during the synthesis process. In addition, a U-Net discriminator with spectral normalization improves discriminator capability and stabilizing training. Some of the the-state-of-the-art methods [11,12] propose using not only a single generator, but also multiple generators in GAN. Specifically, various generators were combined as a mixture of the probabilistic model, and one was selected as the best generator to yield the final output. 

A feature pyramid network (FPN) [15] is a feature extractor that takes a single-scale image of any size as the input and outputs proportionally sized feature maps at multiple levels in a fully convolutional manner. This process is separate from the backbone convolutional architectures. Therefore, the FPN can be used as a generic solution for building feature pyramids inside deep convolutional networks for tasks like object detection [19,33,34,38,39,40]. The construction of the pyramid applies a bottom-up pathway and a top-down pathway. Specifically, the bottom-up pathway is the backbone ConvNet’s feedforward computation, which computes a feature hierarchy consisting of feature maps at various scales with a scaling step. On the other hand, higher resolution features are produced by the top-down pathway, which upsamples spatially coarser but semantically stronger feature maps from higher pyramid levels. These features are then enhanced with features from the bottom-up pathway through lateral connections. Each lateral connection merges feature maps of the same spatial size from the bottom-up and top-down pathways. For instance, the R-CNN family, such as Faster R-CNN [19] and Cascade R-CNN [33], as well as free anchor methods like Foveabox [39], always use the FPN inside their structures. The YOLO Family, including YOLOv4 [34], YOLOv5 [41], and YOLOX [38], as well as the most recent pyramid vision transformer or PVTv2 [40], utilized enhancement of feature pyramid networks as the primary structural component. Moreover, the latest feature pyramid model, DetectoRS [18], proposed recursive feature pyramid network (RFP) to incorporate extra feedback connections from Feature Pyramid Networks into the bottom-up backbone layers, which significantly improves object detection performance.

To hypothesize object locations, state-of-the-art object detection networks [18,19,33,39] rely on region proposal algorithms. A region proposal network (RPN) [19] shares full-image convolutional features with the detection network, allowing for near-free region proposals. It is a fully convolutional network that predicts object bounds and objectness scores at each position at the same time. The RPN was trained inside the end-to-end network to generate high-quality region proposals. Furthermore, Side-Aware Boundary Localization (SABL) [20] was proposed for precise localization proposals in object detection. Each side of the bounding box is localized with a dedicated network branch. Empirically, this method observes that when they manually annotate a bounding box for an object, it is frequently easier to align each side of the box to the object boundary than to move the box while tuning the size. In response to this observation, SABL positions each side of the bounding box based on its surrounding context.

Currently, deep learning-based methods have been widely adopted in many application tasks, including the development of vision-based inspection systems. For instance, the work in ref. [42] applied deep learning convolutional neural networks to determine defect areas on highway roads. Furthermore, the proposed system in ref. [43] used deep learning object detection to develop a defect inspection system for highway roads. Moreover, object detection for defect detection systems (detector) has been widely applied in the manufacturing process, including the steel industry [21,22,23,24,25,26]. Feature pyramid networks (FPN) and region proposal network (RPN) are the main detector components in these previous works. Specifically, the proposed approach in refs. [21,22] utilized FPN inside of a YOLO structure to extract multi-scale features from images of steel and aircraft products. The proposed framework in [23,24,25,26] also adopted FPN to acquire multi-scale features from images of steel, wood, solar cells, and electrical products combined with RPN to localize the defect regions accurately.

The proposed EnsGAN-SDD is mainly inspired by the literature reviewed above to construct the steel surface inspection system. First, ESRGAN was selected as the input image enhancement technique rather than Real ESRGAN since Real ESRGAN produces more noise when facing lower resolutions. Moreover, we proposed a novel EnsGAN that adopted the concept of the multi-Generator GAN to improve the performance of the baseline method in terms of accuracy and processing speed. Then, in the detection stage or steel defect detector (SDD), Side-Aware Boundary Localization (SABL) was integrated into the latest state-of-the-art feature pyramid network, or DetectoRS. The experiments on the common steel surface dataset and Severstal steel dataset showed the proposed work on top of the residual network, and the aggregate residual network backbone achieved significant improvement over the state-of-the-art object detection methods [19,33,34,38,39,40]. Moreover, the proposed EnsGAN SDD also met the criteria of inference time [44,45] in the steel surface inspection system, making it suitable for real-world scenarios in the steel industry.

## 3. Proposed Methods

We proposed two processes for detecting steel defect areas: preprocessing and feature extraction, as shown in Figure 1. The preprocessing stage in our method, named the EnsGAN model, is integrated with the state of the art recursive feature pyramid network (DetectoRS) [18] on top of a residual network (ResNet) [46] or residual aggregation network (ResNext) [47], with Boundary Localization [20] as the localization and recognition network in the feature extraction stage. Moreover, in this work, we aimed to improve super resolution quality from the Enhanced Super-Resolution Generative Adversarial Network (ESRGAN) and reduce the latency of process inference for the steel image enhancement method. This section introduces the base architectures of the proposed EnsGAN (generator and discriminator) used in this research and our strategy to improve quality and reduce latency for the steel defect problem. The resulting image from the EnsGAN is used as the input for DetectorRS with the SABL prediction head [20] as the steel defect detector (SDD) to produce a highly accurate defect localization system for steel surface images.

In the first step of the preprocessing procedure, the discriminator should be trained to classify an image as “High Resolution” or “Low Resolution”. The generator will train to generate the super-resolution image and test it in the discriminator. The loss and information from the discriminator will be used in the generator to enhance the weight to create better resolution. For further detail, the Algorithm 1 shows the complete process flow from preprocessing through detection:
**Algorithm 1** Algorithm of the proposed method Procedure EnsGAN:   Train discriminator (VGG28) with HR and LR to identify HR/LR   Split each LR image into n pieces (n = 2, 4, 8, …)   Initiate n generator (RRDB)   For i = 1 to iteration do    Generate HR with generator with each piece of LR image    Combine all result generators into 1 image according to positions    Check whether X_generator_ is HR or LR using the discriminator    If Y_discriminator_ = 1 do     Pass    Else do     Update weight and bias in generator     Repeat training generator Procedure Feature Extraction:   Convolution input image with RestNet-50   Localize enhancement using recursive feature pyramid (RFP)   Boundary localization using SABL Prepare the data images, convert HR to LR (use bicubic algorithm) Prepare dataset in Pascal VOC format Run Procedure EnsGAN Run Procedure Feature Extraction

### 3.1. Proposed EnsGAN

Our base model uses ESRGAN [9], which achieves better quality than previous super-resolution models such as SRCNN [48], EDSR [49], SRGAN [36], and RCAN [50]. The ESRGAN model is an improvement on the model from SRGAN that removes batch normalization (BN) in the generator structure and uses the residual in residual dense block (RRDB), as depicted in Figure 2.

Removing batch normalization [49] can reduce memory usage by up to four times, achieve better performance than the standard ResNet structure model, and serve as the best simple strategy for sharpening and deblurring images [51]. In the generator structure, ESRGAN keeps the high-level SRGAN architecture. The main difference is that each block contains some convolution layer and activation functions using Leaky Rectified Linear Unit (LReLU) to prevent dead nodes [9]. Improvement can be seen by scaling down the residuals by multiplying a constant between 0 and 1 before adding them to the main path to prevent instability and using initialization when training has a slight variance.

For the discriminator, ESRGAN uses a relativistic discriminator to approximate the probability of an image being real or fake. A generator uses a linear combination of perceptual difference between real and fake images using the VGG-28 network architecture, pixel-wise absolute difference between real and fake images, and the relativistic average loss between real and fake images function during adversarial training. In standard discriminator GAN, the discriminator can be defined as:(1)$$\left(x\right)=\mathrm{Sigmoid}\left(R\left(x\right)\right)\approx \;\sigma \left(R\left(x\right)\right)$$
where $R\left(x\right)$ is a non-transformed layer, and $D\left(x\right)$ is a discriminator [36]. A relative discriminator can be represented as:(2)$$D\left(x\_\mathrm{real},x\_\mathrm{fake}\right)=\sigma \left(R\left(x\_\mathrm{real}\right)-\;E\left[R\left(x\_\mathrm{fake}\right)\right]\right)$$
(3)$$D\left(x\_\mathrm{fake},x\_\mathrm{real}\right)=\sigma \left(R\left(x\_\mathrm{fake}\right)-\;E\_\mathrm{real}\;\left[R\left(x\_\mathrm{real}\right)\right]\right)$$
where E represents the operation of taking the average of the data in the mini-batch. The discriminator loss is then defined as:(4)$$L_{D}^{\mathrm{Ra}}=-E_{x_{\mathrm{real}}}\left[\log\left(D_{\mathrm{Ra}}\left(x_{\mathrm{real}}\right)\right)\right]-E_{x_{\mathrm{fake}}}\left[\log\left(1-{\;D}_{\mathrm{Ra}}\left(x_{\mathrm{fake}}\right)\right)\right]$$

And the adversarial loss for a generator in the symmetrical form can be written as:(5)$$L_{G}^{\mathrm{Ra}}=-E_{x_{\mathrm{real}}}\left[\log\left(1-{\;D}_{\mathrm{Ra}}\left(x_{\mathrm{real}}\right)\right)\right]-E_{x_{\mathrm{fake}}}\left[\log\left(D_{\mathrm{Ra}}\left(x_{\mathrm{fake}}\right)\right)\right]$$
where:(6)$$D_{\mathrm{Ra}}\left(x\right)\left\{\begin{array}{c} \mathrm{sigmoid}\left(R\left(x\right)-E_{x_{\mathrm{fake}}}R\left(x_{\mathrm{fake}}\right)\right)\;\;\;\mathrm{if}\;x\;\mathrm{is}\;\mathrm{real}\\ \mathrm{sigmoid}\left(R\left(x\right)-E_{x_{\mathrm{real}}}R\left(x_{\mathrm{real}}\right)\right)\;\;\;\;\mathrm{if}\;x\;\mathrm{is}\;\mathrm{fake} \end{array}\right.$$

We present our model to achieve better results and time consumption for steel defect images. Our idea is to use an ensemble multi-Generator rather than a single one in the general ESRGAN model. The input data used for training are a part of the split image, as shown in Figure 3. Specifically, we used a split array image to implement the splitting operation. Next, the proposed EnsGAN inputs the generator using a split low-resolution image, as shown in Figure 4.

### 3.2. Detection Baseline 

In the proposed steel defect detector (SDD), we utilized DetectoRS as a detection baseline. This complex deep learning model has two main cores: recursive feature pyramid (RFP) and switchable atrous convolution (SAC). The RFP in DetectoRS, as shown in Figure 5, was placed at the micro-level, which incorporates extra feedback connections in bottom-up backbone layers, and SAC was placed at the macro-level, which convolves the features with different atrous rates and gathers the result using switch functions.

In feature pyramid networks, the output feature $\delta_{i}$ can be defined by:(7)$$\delta_{i}=\alpha_{i}\left(\delta_{i+1},x_{i}\right),\;\mathrm{where}\;x_{i}=\beta_{i}\left(x_{i-1}\right),\;$$
where $\beta_{i}$ denotes the i-th stage of the bottom-up backbone, and $\alpha_{i}$ denotes the i-th top-down FPN operation. The backbone, equipped with FPN, outputs a set of feature maps $\left\{\delta_{i}\;|\;i\;=1,\;\ldots ,\;S\right\}$, where S is the number of the stages, $x_{0}$ is the input image, and $\delta_{s+1}=0$. After adding the recursive feature pyramid (RFP) for feedback connections, the output feature $\delta_{i}$ will be defined by:(8)$$\delta_{i}=\alpha_{i}\left(\delta_{i+1},x_{i}\right),\;\mathrm{where}\;x_{i}=\delta_{i}\left(x_{i-1},\vartheta_{i}\left(\delta_{i}\right)\right),$$
where $\vartheta_{i}$ denotes the feature transformations before connecting them back to the bottom-up backbone. To implement the recursive operation, we unroll it to a sequential network, i.e., ∀i =1,…, S; t =1,… T.
(9)$$\delta_{i}^{t}=\alpha_{i}^{t}\left(\delta_{i+1}^{t},x_{i}^{t}\right),\;\mathrm{where}\;x_{i}^{t}=\beta_{i}^{t}\left(x_{i-1}^{t},\vartheta_{i}^{t}\left(\delta_{i}^{t-1}\right)\right).$$
where T is the number of unrolled iterations, and superscript t is used to denote operations and features at the unrolled step t. The detection baseline using DetectoRS with ResNeXt backbone, ResNext, performs very well in extracting the deeper feature information. Moreover, the RFP has a robust network because the output of the FPN is brought back to each stage of the bottom-up backbone thought-feedback connection, which means there is a double reading of the feature information from the defect image.

### 3.3. Boundary Localization

To improve the defect localization accuracy, we add Side-Aware Boundary Localization (SABL) [20] to replace the prediction head on the DetectoRS. SABL is a methodology for precise localization in object detection where each side of the bounding box is respectively localized with a dedicated network branch. There is a two-step localization scheme that first predicts a range of movement through bucket prediction and then pinpoints the precise position within the predicted bucket. Figure 6 shows that it first looks for the correct bucket, i.e., the one in which the boundary is located. Fine regression is then performed by predicting offsets using the selected bucket’s centerline as a coarse estimate. This scheme enables very precise localization even in the presence of large variances in displacements. 

## 4. Experiments

For the experiments, we used the common steel surface defect dataset provided by Kaggle, which consists of 12,568 images. This dataset has been used for the real industrial application scenarios [52,53,54]. According to the original source (https://www.kaggle.com/c/severstal-steel-defect-detection, accessed on 24 April 2022), this dataset has four class categories: Class 1, Class 2, Class 3, and Class 4, respectively. Specifically, the study in ref. [54] explained that Class 1 has pitted surface defect conditions, Class 2 has crazing defect conditions, Class 3 has scratch defect conditions, and Class 4 has patch defect conditions. A clear presentation of the defect conditions can be found in Figure 7. Moreover, we trained the dataset using an Nvidia RTX 3090 Ti 24GB with a splitting ratio of 80% of images for the training, 10% for validation, and 10% for the testing. Originally, this dataset used pixel-wise annotation labels. However, in this experiment, we converted the label into a bounding box using the Pascal VOC annotation format [55] to reduce the annotation cost and make the system more suitable for real-world scenarios.

Specifically, in the generator of the proposed preprocessing EnsGAN, for RRDB Network, we used 2D Convolution with a 3 × 3 kernel size and 1 × 1 strides with the same padding, and 2D Transpose Convolution with a 3 × 3 kernel size and 2 × 2 strides with the same padding; for the activation function, we used Leaky ReLU with α = 0.2. Furthermore, in VGG28 Architecture, we used Convolution 2D with a 3 × 3 kernel size with the same padding, Leaky ReLU activation function with α = 0.2, and fully connected layers with 100 neurons. In the localization and recognition network experiments, we used a single image as the batch size with stochastic gradient descent as the training optimizer. Moreover, the initial learning rate was set to 0.001, and the decay rate step was 0.0001 in epochs 16 and 19 of the 40 epochs. In addition, we only used one general data augmentation method, random flipping, to increase the diversity feature of the input image.

Finally, in this section, we discuss the results of our scope comparison for super resolution with ESRGAN [9] and Real-ESRGAN [10] in the preprocessing stage. Furthermore, to offer a complete view of performance results for the proposed method, we used the state-of-the-art object detection models that are feature pyramid network-based. From the YOLO family, these include: YOLOv4 [34], YOLOv5 [41], YOLOX [38]; from the free anchor method: FoveaBox [39]; from the transformer network: PVTv2 [40]; and from the R-CNN family: Faster R-CNN [19], and Cascade R-CNN [33]; for comparison with our proposed inspection system, EnsGAN-SDD.

### 4.1. Results and Discussion of Proposed Preprocessing

In this subsection, we demonstrate the performance of our proposed method in the preprocessing stage. This experiment tried to sample the best split number ranging from 2 up to 32 based on the Peak Signal-to-noise Ratio (PSNR) to measure the quality between the original and the reconstructed image. We used three generated defect images produced by Real-ESRGAN, ESRGAN, and proposed EnsGAN to perform the quantitative comparison. Moreover, we cropped the specific regions in these three defect samples images and have shown them in Figure 8 as Image 1 on the top, Image 2 in the middle, and Image 3 on the bottom for the qualitative comparison. As shown in Table 1, the objective results of the proposed EnsGAN surpass the quality of the original Real-ESRGAN and ESRGAN with the maximum PSNR numbers on the three images of 42.005, 39.278, and 43.135 when the split ratio was equal to 16. Moreover, Table 2 shows the benefit of our proposed EnsGAN in terms of processing speed efficiency. As the split ratio increased, the processing time simultaneously decreased. As shown in this table, for the three images, Real-ESRGAN and ESRGAN took 11.927, 12.473, 11.638, and 11.687, 11.723, 11.656 s, respectively. Our proposed method with a split ratio of 16 showed highly superior performance of 7.937, 8.250, and 8.123 s. Our proposed method with a split number of 16 improved greatly to 7.937, 8.250, and 8.123 s.

For the qualitative comparison or subjective comparison performance, Figure 8 shows the original sample defect image and the generated defect sample images of results produced by Real-ESRGAN, ESRGAN, and the proposed EnsGAN. The Figure shows that the adversarial image of the proposed method enhanced the resolution quality, especially in the defect region marked by the red and yellow boxes. Moreover, the green box shows that the proposed method has better noise reduction on the original images and attained a smoother texture than ESRGAN did. On the other hand, the output of Real-ESRGAN enhanced the resolution, but unfortunately increased the noise because of aliasing issues. The problem arises because the traditional degradation model, which consists of a blur, downsampling, noise, and JPEG compression, leads to insufficient ability to model real-world degradations.

Based on the discussion above, we conclude that the proposed super-resolution method EnsGAN performs well in all aspects compared with the state-of-the-art methods. As we know, good sample data are very important for the neural network model. Therefore, the quality of the produced image and efficiency of the speed performance from our proposed method can benefit the performance of the steel surface inspection system.

### 4.2. Results and Discussion of Proposed Inspection System

In the experiments on the localization and recognition network highlighted in this section, we used recall (REC), average precision (AP), and mean average precision (mAP) to align with the quantitative evaluation in Pascal VOC object detection [55]. First, Table 3 shows the ablation improvement of the proposed method. We used two state-of-the-art models, known as residual network 50 (ResNet-50), as the light backbone, and residual aggregation network 101 (X-101) as the deeper backbone for the DetectoRS as the defect detector’s baseline. With the first ResNet-50 backbone, the mAP accuracy of the system was 77.1%. When we added EnsGAN to the system, the accuracy increased to 78.8%. After Side-Aware Boundary Localization (SABL) was substituted for the general prediction head, accuracy improved to 79%. Furthermore, we demonstrated the proposed system in a deeper network backbone. The initial mAP accuracy of the DetectorRS on top of X-101 was 78.5%. After we integrated the proposed EnsGAN and SABL, the mAP improved to 79.2% and 80.4%, respectively.

Table 4 further shows the proposed system’s objective performance over the state-of-the-art object detection models. The proposed EnsGAN-SDD on top of ResNet-50 improved the accuracy of the YOLO family: YOLOv4, YOLOv5, and YOLOX, anchor free method: Foveabox; transformer network: PVTv2; and R-CNN family: Faster R-CNN and Cascade R-CNN with a margin of 18.2%, 18.9%, 13.8%, 12.4%, 7.7%, 13.7%, and 11.5%, respectively. Figure 9, in turn, shows how accurate the proposed steel surface inspection system was in localizing four types of each defect class in the Severstal steel defect dataset. Moreover, Figure 10 shows the proposed system’s capabilities in multi-class defect localization. These results show that the proposed method obtained high performance in detecting challenging steel product defects. In addition, the proposed system achieved 11.6 frames per second (FPS) in a single GPU Nvidia RTX 3090 Ti on top of the ResNet-50 backbone. This shows that EnsGAN-SDD can be applied in the steel industry, which requires a processing speed above 10 FPS, as explained in refs. [44,45].

## 5. Conclusions

The proposed EnsGAN-SDD can offer a solution to improve steel surface inspection systems. EnsGAN achieved a good PSNR on the steel defect dataset and had a low processing time relative to ESRGAN and Real-ESRGAN. Moreover, when it was integrated with robust detectors or DetectoRS and Side-Aware Boundary Localization (SABL) as steel defect detectors (SDD), the localization accuracy significantly improved over those of state-of-the-art object detection models while maintaining the necessary processing speed to meet standards for steel surface inspection systems. Furthermore, in the steel defect detection domain, we found that the main challenge lies not only in the lower resolution, but also in the difficulty of examples, e.g., the tiny defect size in Class 1 or pitted surface defect conditions, which have lower accuracy than other classes. Future work must extend the proposed preprocessing stage to address this issue.

## Figures and Tables

**Figure 1 sensors-22-04257-f001:**
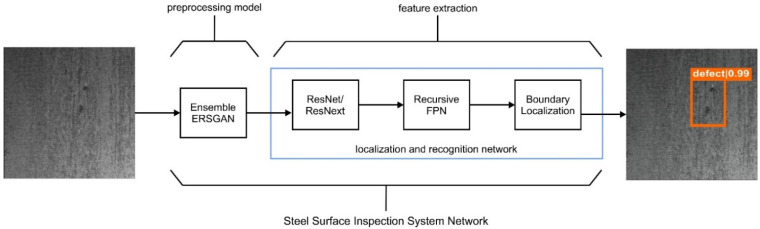
Flow Process detecting steel defects image. In the first stage, we proposed the preprocessing model named EnsGAN to enhance the lower resolution of input image. The next stage is feature extraction that utilizes the combination of DetectoRS, which has Recursive FPN on top of the Residual Network (ResNet) or residual aggregation network (ResNext) backbone, and side aware boundary localization in the prediction head.

**Figure 2 sensors-22-04257-f002:**
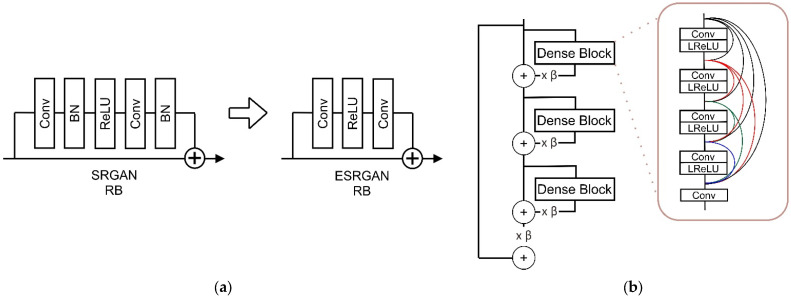
(**a**) ESRGAN improves the SRGAN with removing BN in the residual block. (**b**) The Residual in residual dense block (RRDB) and residual scaling parameter $\left(\beta \right)$ used at the SRGAN structure.

**Figure 3 sensors-22-04257-f003:**
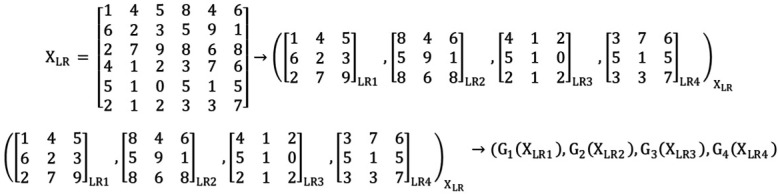
Example of splitting an array image into a four-piece image, then inputting it into Generator EnsGAN.

**Figure 4 sensors-22-04257-f004:**
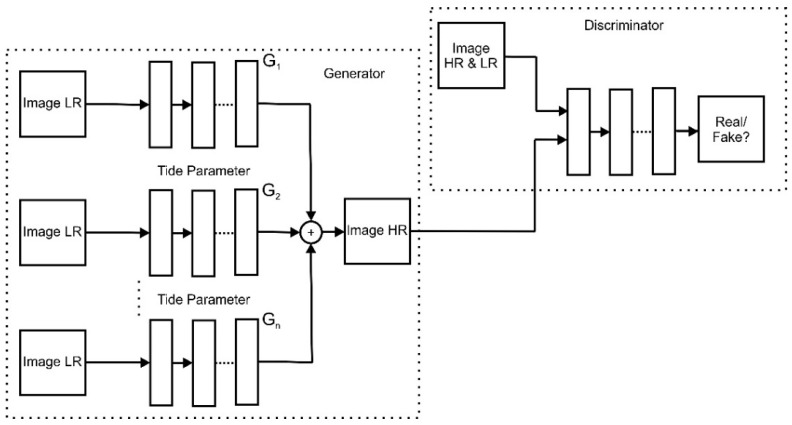
Proposed EnsGAN model with n generators and a discriminator. As illustrated in Figure 3, splitting an array image or Image LR from one single image and inputting it into a multi-Generator $\left(G_{1},{\;G}_{2},\;\ldots ,{\;G}_{n}\right)$ then uses the ensemble technique to reconstruct the output Image HR.

**Figure 5 sensors-22-04257-f005:**
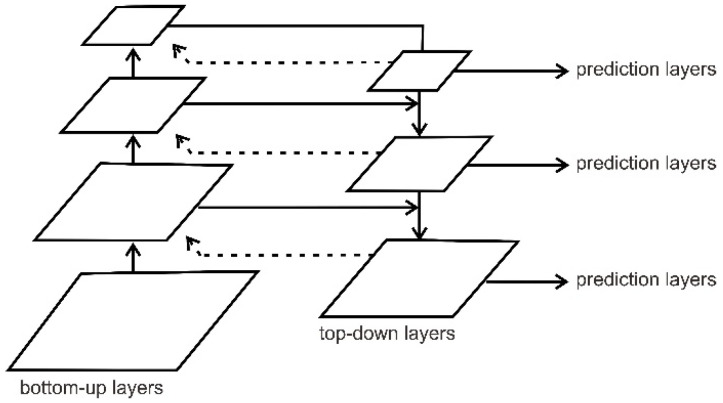
Visualization of recursive feature pyramid networks (RFP). The feedback connections are used to learn more deeply about the features’ information.

**Figure 6 sensors-22-04257-f006:**
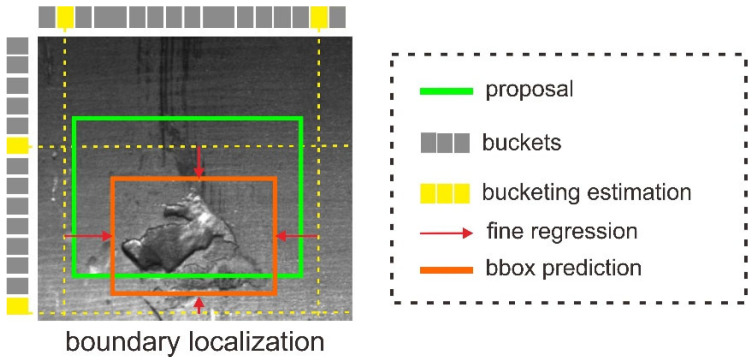
Visualization of Side Aware Boundary Localization (SABL). To replace the proposal: the bucketing estimation is used to predict the buckets candidate and employs the fine regression to acquire the final bounding box (bbox) prediction.

**Figure 7 sensors-22-04257-f007:**
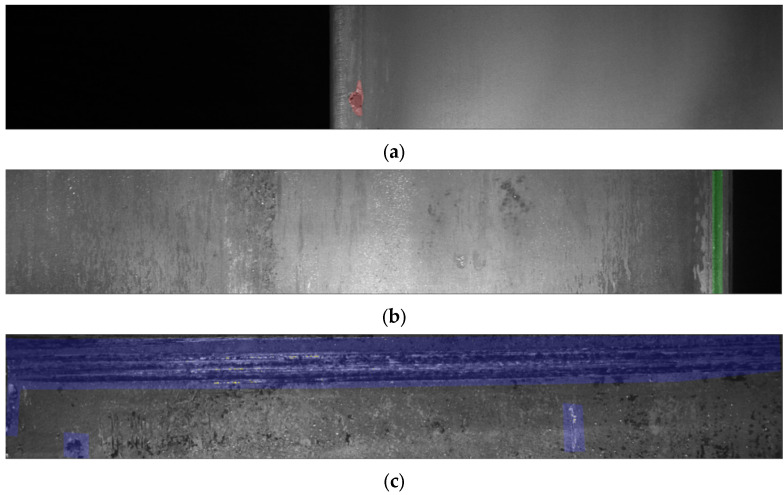
The Severstal steel defect dataset has four defect classes marked with different colors. (**a**) Class 1 has the pitted surface defect condition; (**b**) class 2 has the crazing defect condition; (**c**) class 3 has the scratches defect condition; and (**d**) class 4 has the patches defect condition.

**Figure 8 sensors-22-04257-f008:**
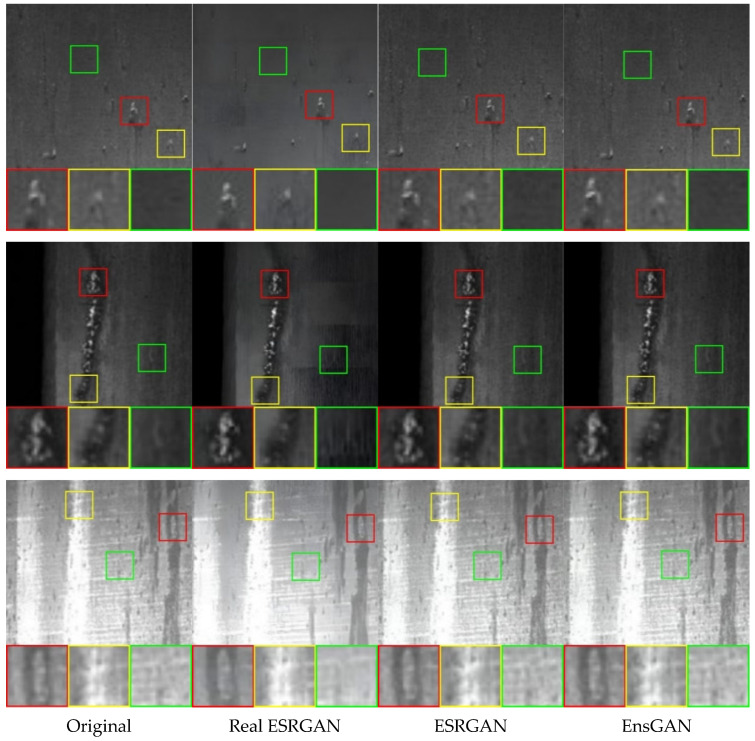
From left to right (zoom in for the best view): the original image and the generated image produced by Real-ESRGAN, ESRGAN, and the proposed EnsGAN, respectively. The red and yellow boxes show that the proposed EnsGAN does a better job of improving the resolution quality of the defect region. Moreover, the green box shows that the proposed method has better noise reduction on the original images and attains a smoother texture than ESRGAN does.

**Figure 9 sensors-22-04257-f009:**
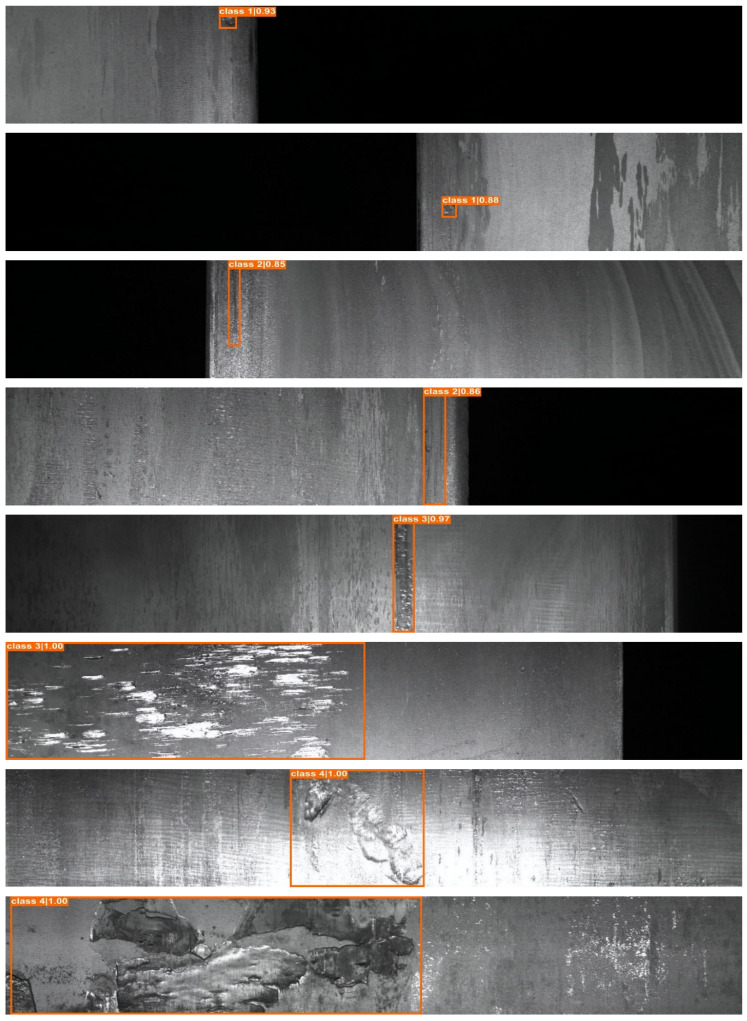
Sample of defect localization results for each class of the Severstal dataset.

**Figure 10 sensors-22-04257-f010:**
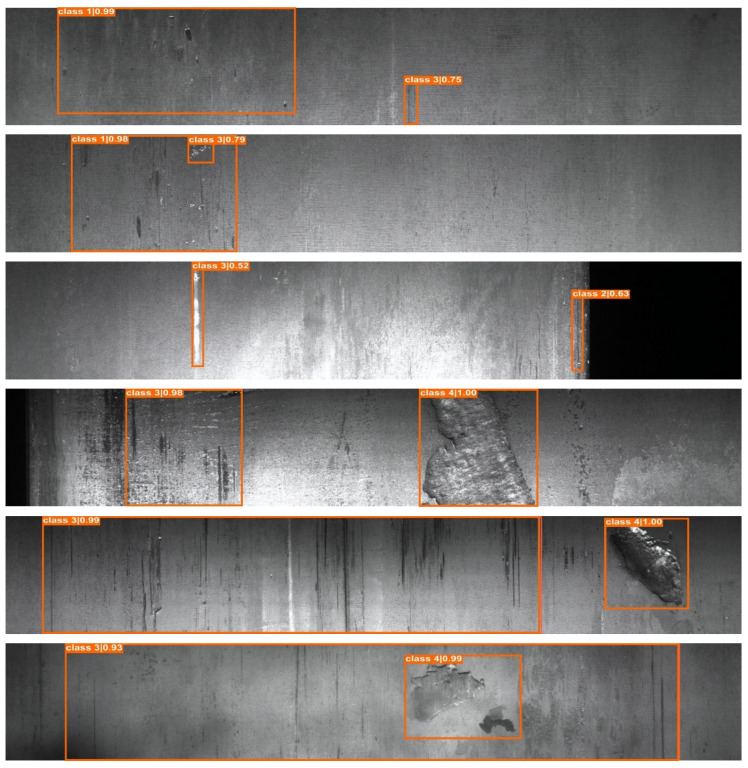
Sample of multi-class defects localization results of the Severstal steel dataset.

**Table 1 sensors-22-04257-t001:** Comparison of Real-ESRGAN, ESRGAN, and proposed EnsGAN for PSNR results on three sample images from the Severstal steel defect dataset. The defect samples in these three images are shown in Figure 8 in the following order: the top is Image 1, the middle is Image 2, and the bottom is Image 3.

**Input Image**	**Real-ESRGAN**	**ESRGAN**	**EnsGAN (2×)**
Image 1	21.476	41.895	41.932
Image 2	28.174	39.205	39.225
Image 3	29.539	43.048	43.079
**Input Image**	**Real-ESRGAN**	**ESRGAN**	**EnsGAN (4×)**
Image 1	21.476	41.895	41,949
Image 2	28.174	39.205	39.233
Image 3	29.539	43.048	43.081
**Input Image**	**Real-ESRGAN**	**ESRGAN**	**EnsGAN (8×)**
Image 1	21.476	41.895	41.953
Image 2	28.174	39.205	39.254
Image 3	29.539	43.048	43.083
**Input Image**	**Real-ESRGAN**	**ESRGAN**	**EnsGAN (16×)**
Image 1	21.476	41.895	42.005
Image 2	28.174	39.205	39.278
Image 3	29.539	43.048	43.135
**Input Image**	**Real-ESRGAN**	**ESRGAN**	**EnsGAN (32×)**
Image 1	21.476	41.895	41.789
Image 2	28.174	39.205	39.182
Image 3	29.539	43.048	42.975

**Table 2 sensors-22-04257-t002:** Comparison of EnsGAN, ESRGAN, and Real-ESRGAN for speed process on Severstal Steel defect dataset. The defect samples of these three images are shown in Figure 8 in the following order: the top is image 1, the middle is image 2, and the bottom is image 3.

**Input Image**	**Real-ESRGAN**	**ESRGAN**	**EnsGAN (2×)**
Image 1	11.927s	11.687s	11.210s
Image 2	12.473s	11.723s	11.137s
Image 3	11.638s	11.656s	11.199s
**Input Image**	**Real-ESRGAN**	**ESRGAN**	**EnsGAN (4×)**
Image 1	11.927s	11.687s	10.548s
Image 2	12.473s	11.723s	10.497s
Image 3	11.638s	11.656s	10.511s
**Input Image**	**Real-ESRGAN**	**ESRGAN**	**EnsGAN (8×)**
Image 1	11.927s	11.687s	9.572s
Image 2	12.473s	11.723s	9.611s
Image 3	11.638s	11.656s	9.557s
**Input Image**	**Real-ESRGAN**	**ESRGAN**	**EnsGAN (16×)**
Image 1	11.927s	11.687s	7.937s
Image 2	12.473s	11.723s	8.250s
Image 3	11.638s	11.656s	8.123s
**Input Image**	**Real-ESRGAN**	**ESRGAN**	**EnsGAN (32×)**
Image 1	11.927s	11.687s	6.950s
Image 2	12.473s	11.723s	6.944s
Image 3	11.638s	11.656s	6.877s

**Table 3 sensors-22-04257-t003:** The ablation improvement results of the proposed EnsGAN-SDD.

Improvement	REC-1	REC-2	REC-3	REC-4	AP-1	AP-2	AP-3	AP-4	mAP
Baseline R-50	0.814	0.800	0.878	0.922	0.663	0.727	0.806	0.890	0.771
+ EnsGAN	0.791	0.850	0.883	0.953	0.647	0.807	0.799	0.898	0.788
+ SABL	0.791	0.850	0.863	0.969	0.668	0.813	0.775	0.904	0.790
Baseline X-101	0.802	0.850	0.883	0.984	0.653	0.765	0.803	0.922	0.785
+ EnsGAN	0.826	0.850	0.888	0.938	0.705	0.774	0.812	0.876	0.792
+ SABL	0.849	0.850	0.883	0.922	0.727	0.801	0.800	0.889	0.804

**Table 4 sensors-22-04257-t004:** Comparison results of the proposed EnsGAN-SDD with the state-of-the-art object detection models.

Model	Baseline	REC-1	REC-2	REC-3	REC-4	AP-1	AP-2	AP-3	AP-4	mAP
YOLOv4	CSPDarknet	0.895	0.850	0.949	0.922	0.443	0.615	0.664	0.710	0.608
YOLOv5	CSPDarknet	0.837	0.900	0.921	0.906	0.423	0.693	0.607	0.682	0.601
YOLOX	CSPDarknet	0.802	0.900	0.921	0.828	0.529	0.628	0.745	0.707	0.652
FoveaBox	R-50	0.826	0.750	0.905	0.953	0.574	0.450	0.771	0.870	0.666
PVTv2	PVTv2-B5	0.895	0.850	0.934	0.953	0.579	0.637	0.779	0.856	0.713
Faster RCNN	R-50	0.756	0.850	0.879	0.891	0.531	0.746	0.636	0.700	0.653
Cascade RCNN	R-50	0.802	0.850	0.878	0.891	0.608	0.714	0.684	0.696	0.675
EnsGAN-SDD (our)	R-50	0.791	0.850	0.863	0.969	0.668	0.813	0.775	0.904	0.790
EnsGAN-SDD (our)	X-101	0.849	0.850	0.883	0.922	0.727	0.801	0.800	0.889	0.804

## Data Availability

Author’s datasets. Available online: https://www.kaggle.com/competitions/severstal-steel-defect-detection/data (accessed on 24 April 2022).

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
