# Peer review of "Enhancing Precision with an Ensemble Generative Adversarial Network for Steel Surface Defect Detectors (EnsGAN-SDD)"

_sensors, 2022, doi:10.3390/s22114257_

Round 1

Reviewer 1 Report

1 in addition to the visual display of the data generated by GAN, can the author give a detailed comparison of quantitative parameters

Author Response

Dear Reviewer,

Thank you very much for your thoughtful review of our manuscript. We appreciate the time and effort that the reviewers have dedicated to providing valuable feedback, and are grateful for their insightful comments on this paper. We have been able to incorporate changes to reflect Your suggestions, and have highlighted the changes within the manuscript. Moreover, please find details in the attached file.

Reviewer 2 Report

In this paper, an algorithm named EnsGAN for steel surface defect detectors is proposed. The proposed EnsGAN method combined ESRGAN, Recursive FPN and Boundary Localization to improve the performance of defect detectors and shorten the processing time. The method is verified theoretically and experimentally. This paper is logical and innovative, but the writing is not well, and there are some deficiencies in details.
1、 In Abstract and Figure 1, the use of ERSGAN as an image enhancement technology in the pre-processing stage is mentioned, but in this paper, ESRGAN is often mentioned in the pre-processing stage in many places. ERSGAN is inconsistent with ESRGAN. Please check whether it is a spelling error.
2、 It is mentioned in Abstract that "Next, in the detector section, recursive feature pyramid network is utilized to extract deeper multi-scale steel features by learning the feedback from the sequential feature pyramid network", and it’s mentioned in Section 2 that "in the detection stage or steel defect detector (SDD), Side-Aware Boundary Localization (SABL) is integrated into the latest state-of-the-art feature pyramid network, or DetectoRS". The description of the RFP and DetectoRS is unclear and misleading. Please provide supplementary information for DetectoRS.
3、 It is mentioned in Section 4 that "which consists of 12,568 images including four defect classes", What are the four defect classes? Please add some details.
4、 It is mentioned in Section 4.1 that " The figure shows that the adversarial image of the proposed method enhanced the resolution quality, reduced the noise from the original images, and attained a smoother texture than ESRGAN did", However, it is difficult to observe from Figure 8 that the proposed EnsGAN method can obtain a smoother texture than ESRGAN.
5、 As shown in Figure 9, the detection results of four defects on steel surface are shown respectively. However, if multiple defects exist on steel surface at the same time, can the proposed EnsGAN method still accurately recognize and locate the defects?

Author Response

Dear Reviewer,

Thank you very much for your thoughtful review of our manuscript. We appreciate the time and effort that you have dedicated to providing valuable feedback. We are grateful to the reviewers for their insightful comments on this paper. We have been able to incorporate changes to reflect the suggestions provided. We have highlighted the changes within the manuscript. Specifically, we have carefully revised the unclear definitions in the earlier version of the manuscript. Moreover, please find details in the attached file.

Reviewer 3 Report

In this paper, GAN-type DNN-s are applied to the problem of steel surface defect detection. Overall, the manuscript is nicely written, but there are still a few concerns, some of them major.

First, the literature review could cover more works from the AI based defect detection domain. Numerous papers have been published on this topic and cross-referencing various methodologies for defect detection could be of potential use to your readership. For instance, you can consider reviewing some research works such as [https://doi.org/10.3390/s22093537], [https://doi.org/10.3390/s22093467], [https://doi.org/10.3390/app9224829], [https://doi.org/10.3390/s22093417] to give better context to the defect detection problem, in general.

For the GAN-based approaches, it would actually be best, if you provided a tabular summary of existing methodologies with references to the most relevant research. That way, it would be easier to position your contribution against the state of the art and identified gaps.

This leads to the next point, which is lack of clear statement of contribution. In the introduction section, there is an attempt at it near line 86, but the referee would appreciate a more clear statement including statement of novelty, perhaps in a separate paragraph with a heading. Granted, in the manuscript the authors developed some methods to enhance the GAN architecture, but just combining some NN subarchitectures together (e.g., combining into an ensemble) could be considered a weak scientific contribution. Especially since it is only verified on a benchmark dataset and not applied in a real industrial application (which would then make the contribution much stronger). Hence, please provide a better motivation for your contribution and describe its novelty and potential in more details.

Please expand the conclusions section. You only mentioned some of the benefits of the proposed solution, but it would be nice to hear about some of the drawbacks to encourage further work, which you should also describe in the same section.

Please be more specific in figure captions as well. For instance, just looking at Figure 8 does not tell the reader much at first glance. It is described at line 322, but it would be beneficial if just looking at the figure and the caption, one would understand what the figure conveys.

There are two sections numbered 4.1 in the document. Please fix it. Additionally, please check ref. [14] as the link there is broken.

Concerning formatting and typesetting. Images containing diagrams, such as in Figs. 1 and 2, should either be of higher resolution (above 300dpi), or the schematic parts should be vector-based images.

The use of language is mostly fine. Some proofreading could be performed as usual following the revision of the initial submission.

It is recommended the article be sent back to authors for a major revision.

Author Response

Dear Reviewer,

Thank you very much for your thoughtful review of our manuscript. We appreciate the time and effort that the reviewer has dedicated to providing your valuable feedback on our manuscript. We are grateful to the reviewers for their insightful comments on this paper. We have been able to incorporate changes to reflect the suggestions provided by the reviewer. We have highlighted the changes within the manuscript. Specifically, we have carefully revised several concerns in the early version of the manuscript. Moreover, please find details in the attached file.

Round 2

Reviewer 2 Report

The authors have revised the manuscript according to the comments, which can be published after text editing and revising.

Reviewer 3 Report

Thank you for addressing my comments.